# A Case of Pneumothorax Ex Vacuo Associated with COVID-19

**DOI:** 10.3390/medicina59040709

**Published:** 2023-04-04

**Authors:** Ryusei Yoshino, Nana Yoshida, Shunsuke Yasuda, Akane Ito, Masaki Nakatsubo, Masahiro Kitada

**Affiliations:** Department of Thoracic Surgery and Breast Surgery, Asahikawa Medical University Hospital, 2-1-1-1 Midorigaoka Higashi, Asahikawa-shi 078-8510, Japan; nana1524@icloud.com (N.Y.); s-yasuda@asahikawa-med.ac.jp (S.Y.); itoakane@asahikawa-med.ac.jp (A.I.); tsubo528@asahikawa-med.ac.jp (M.N.); k1111@asahikawa-med.ac.jp (M.K.)

**Keywords:** pneumothorax ex vacuo, coronavirus disease, pneumothorax, mediastinal emphysema, subcutaneous emphysema

## Abstract

Pneumothorax is a known complication of coronavirus disease 2019 (COVID-19). The concept of pneumothorax ex vacuo has also been proposed to describe pneumothorax that occurs after malignant pleural effusion drainage. Herein, we present the case of a 67-year-old woman who had abdominal distension for 2 months. A detailed examination led to the suspicion of an ovarian tumor and revealed an accumulation of pleural effusion and ascitic fluid. Thoracentesis was performed, raising the suspicion of metastasis of high-grade serous carcinoma arising from the ovary. An ovarian biopsy was scheduled to select subsequent pharmacotherapy, and a drain was inserted preoperatively into the left thoracic cavity. Thereafter, a polymerase chain reaction analysis revealed that the patient was positive for COVID-19. Thus, the surgery was postponed. After the thoracic cavity drain was removed, pneumothorax occurred, and mediastinal and subcutaneous emphysema was observed. Thoracic cavity drains were then placed again. The patient’s condition was conservatively relieved without surgery. This patient may have developed pneumothorax ex vacuo during the course of a COVID-19 infection. Since chronic inflammation in the thoracic cavity is involved in the onset of pneumothorax ex vacuo, careful consideration is required for the thoracic cavity drainage of malignant pleural effusion and other fluid retention.

## 1. Introduction

Pneumothorax is a known complication of coronavirus disease 2019 (COVID-19). At the beginning of the COVID-19 epidemic, pneumothorax due to pressure injury associated with ventilator use was frequently reported; however, since then, pneumothorax has been reported to occur in COVID-19 patients even without ventilator use. [1]. Symptoms of pneumothorax include chest pain and shortness of breath, and a chest X-ray is commonly used for diagnosis. Pneumothorax typically requires immediate diagnosis and intervention such as thoracic drainage; if missed, a tension pneumothorax may develop, leading to cardiopulmonary decompensation and death in the most severe cases. On the contrary, small spontaneous pneumothoraces usually heal without treatment and only require observation [2].

The concept of pneumothorax ex vacuo has also been proposed to describe pneumothorax that occurs after drainage of malignant pleural effusion [3]. The mechanism of occurrence is not clear, but it is believed to result from the formation of a fibrous pleura due to the infiltration of malignant cells, which limits lung re-expansion. This results in negative pleural pressure and the suction of tissue air into the pleural space, in a phenomenon similar to acute lobar collapse. Much controversy exists regarding the treatment of pneumothorax ex vacuo. However, when the cause is malignant pleural effusion, the prognosis is often poor; thus, conservative therapy rather than active surgical intervention is often performed [2].

Recently, there have been scattered reports of complications after COVID-19; however, several details remain unknown. Herein, we report a case in which a patient developed pneumothorax ex vacuo during the course of a COVID-19 infection. To date, there have been no similar case studies published.

## 2. Case Presentation

A 67-year-old woman presented to the hospital with a chief complaint of abdominal distension, which had persisted for 2 months. No other symptoms such as respiratory distress or chest pain were noted. Regarding her past medical history, she had hypertension and had undergone surgery (craniotomy intracranial lumpectomy) for cerebellar hemangioblastoma. There was nothing noteworthy regarding the patient’s family or smoking history.

Upon admission, the patient’s height was 149 cm, her weight was 56.0 kg, and her body mass index was 25.2 kg/m^2^. Moreover, neither an anemic eyeball nor conjunctival anemia was observed. No lymph nodes on the body surface were palpable. Heart sounds were normal, but lung sounds were mildly diminished on the left. The abdomen was distended and soft, without tenderness. There was no edema in the bilateral lower legs.

Regarding the laboratory findings on admission, no abnormal findings were noted in the urinalysis, blood count, biochemical examination, or coagulation testing. Her tumor marker (CA-125) was elevated to 629 U/mL. Further, there were no abnormal findings from the electrocardiogram or respiratory function tests.

A computed tomography (CT) scan performed for close examination of abdominal distention revealed pleural and ascites effusions, which led to suspicion of a neoplastic lesion arising from the ovaries (Figure 1).

Thoracentesis and abdominocentesis were performed for detailed examination, and 1000 mL of each specimen was obtained. The cell block results of the obtained specimens indicated a suspected metastasis of high-grade serous carcinoma arising from the ovary. A biopsy under general anesthesia was scheduled to confirm ovarian cancer and to select drug therapy. However, after draining 1000 mL of pleural fluid, a large amount of left pleural fluid was still reaccumulated at the second examination two weeks later. Therefore, intubation under general anesthesia was judged to be high-risk. Therefore, the anesthesiologist requested the insertion of a left thoracic drain the day before. After the insertion of a thoracic drain, the patient was found to be COVID-19-positive on preoperative screening, and as a result, the patient’s biopsy surgery was postponed and the left thoracic drain was planned to be removed. However, considering that removing the drain immediately following insertion would be detrimental to the patient, removal was scheduled for the next day, to be followed by observation of the patient for several additional days. As planned, the drain was removed the day after insertion. However, approximately one hour following the drain’s removal, mediastinal and subcutaneous emphysema suddenly developed, with the patient also complaining of mild dyspnea. A left pneumothorax was then diagnosed using chest radiography. In addition, chest radiographic findings included mediastinal and subcutaneous emphysema observed in the anterior mediastinum and on the left side of the chest. Subsequently, a drain (20 Fr) was placed in the patient’s left thoracic cavity immediately after confirming the X-ray (Figure 2).

After insertion of the thoracic drain, a chest CT was performed; the scan showed mediastinal emphysema in the anterior mediastinum, with predominant subcutaneous emphysema in the left chest (Figure 3a,b).

The following day, the subcutaneous emphysema showed a trend of enlargement on the chest X-ray image. Therefore, an additional drain (20 Fr) was placed from the left anterior thoracic region into the left thoracic cavity (Figure 4).

We were able to confirm that the thoracic drain position was not a problem, and neither mediastinal nor subcutaneous emphysema worsened thereafter (Figure 5a,b).

No air leaks were observed; however, respiratory fluctuations were observed. Day by day, the subcutaneous emphysema showed a tendency to improve. One drain was removed on day 9 following drain insertion and the other drain on day 13. No worsening of subcutaneous emphysema was observed after drain removal. The patient was discharged on day 4 after drain removal (Figure 6). 

The final histopathologic examination of the effusion revealed a metastatic high-grade serous carcinoma. Additionally, the cell block specimen showed a small number of small papillary and discrete tumor cells against a background of numerous macrophages. Immunostaining showed that the tumor cells were CK7, PAX8, WT-1, *p*53, and *p*16 diffusely positive; ER partially positive; and CK20, TTF-1, Napsin A, and PgR negative. The histopathology of the ascites fluid showed metastatic adenocarcinoma. The cell block specimen showed a small number of atypical cells in a background of numerous erythrocytes, histiocytes, lymphocytes, neutrophils, and mesothelial cells. Immunostaining showed that the atypical cells were positive for Claudin4, BerEP4, and PAX8 and negative for D2-40 and calretinin. The patient began treatment for ovarian cancer with paclitaxel/carboplatin plus bevacizumab.

## 3. Discussion

This study presented a case of pneumothorax ex vacuo, defined as pneumothorax caused by failure of the lungs to expand despite negative intrathoracic pressure and pleural rupture after drainage of pleural effusion. Reports of this condition are limited [4]. However, there have been several reports of secondary pneumothorax occurring after the drainage of malignant pleural effusion and radiation. Additionally, tuberculosis, allergic bronchopulmonary aspergillosis, and ventilator-associated pneumonia have been previously reported as etiologies for pneumothorax ex vacuo [2]. Several hypotheses have been proposed for the pathogenesis of pneumothorax ex vacuo [5], one being that underlying pleural diseases restrict the lungs and, in turn, inhibit the re-expansion of the lungs [4]. Another hypothesis is that chronic atelectasis causes the depletion of pulmonary surfactants, thereby inhibiting the re-expansion of the lungs [6]. It is said that pneumothorax ex vacuo is involved in the pathology of non-expandable lung (NEL). NEL is the inability of the lung to fully expand against the chest wall after pleural drainage. Active inflammation such as pneumonia, pleurisy, or malignant pleural tumors can be cited as causes for this condition [7].

Pneumothorax ex vacuo is often difficult to distinguish from iatrogenic pneumothorax. However, in the case of iatrogenic pneumothorax, shortness of breath and fatigue appear as physical findings and are often accompanied by vital changes such as hypotension. Conversely, pneumothorax ex vacuo is often asymptomatic. If pneumothorax occurs after thoracentesis but is asymptomatic, it is necessary to carefully examine the indications for pleural drainage. For this purpose, confirmation of physical findings and chest CT examination is useful [8].

Recent reports have suggested that pneumothorax may occur as a complication of COVID-19 infection in 1% of patients with COVID-19 requiring hospitalization [1]. The pathogenesis of this complication may be associated with cystic degeneration of the lungs, diffuse alveolar damage, and cough [9]. Additionally, there have been reports of pneumothorax after noninvasive positive pressure arousal to improve oxygenation during infection with COVID-19 [10]. Furthermore, pathological findings in the background lungs of patients with a history of COVID-19 include diffuse alveolar damage, indicating that such patients may be at risk of developing delayed pneumothorax [11]. In addition, mortality is reported to be higher in patients with COVID-19 and pneumothorax [12,13]. However, while pneumothorax is a complication of COVID-19, there are reports that pneumothorax is not an independent marker of poor prognosis. Still, aggressive treatment is recommended when possible [1]. In our patient, asymptomatic COVID-19 was detected during the drainage of malignant pleural effusion, and pneumothorax ex vacuo was assumed to have occurred after the removal of the thoracic cavity drain.

Conservative treatment is effective for mediastinal and subcutaneous emphysema, which are complications of pneumothorax ex vacuo. Several therapeutic strategies have been proposed for this purpose. For example, Boland et al. [5] have concluded that aggressive treatment, such as thoracic cavity drainage, is unnecessary. However, some reports recommend the therapeutic effects of thoracic cavity drainage [3]. There is also literature suggesting that conservative treatment with bronchoscopy may be effective. In any case, the need for aggressive therapeutic intervention should be determined by patient condition. Even in cases with a poor prognosis such as this one, early improvement of pneumothorax ex vacuo may lead to the early selection of chemotherapy [2].

In addition, even in conservative management, it is necessary to pay close attention to the occurrence of re expansion pulmonary edema. In this condition, pulmonary edema is thought to occur as a result of the rapid re expansion of the lung after pneumothorax or pleural effusion drainage, resulting in reperfusion of pulmonary blood flow and increased vascular permeability. Multiple factors such as pulmonary microvascular injury caused by the re expansion of collapsed lung, active oxygen due to ischemia-reperfusion injury, multinucleated neutrophil infiltration, and decreases in surfactant have been proposed as mechanisms of re expansion pulmonary edema. Additionally, collapse times of longer than 72 h and high negative pressure (>20 cm H_2_O) during chest drainage are considered high risks. Therefore, even in patients with pneumothorax ex vacuole and COVID-19, it is important to recognize and manage risk factors such as long-term collapse, the size of the pneumothorax, and high negative pressure for drainage [14,15].

In our patient, mediastinal and subcutaneous emphysema occurred after the thoracic cavity drain was removed, and the thoracic cavity drains were inserted for a second time. However, the patient’s condition improved minimally soon after reinsertion, and surgery was considered. Since she was at high risk because of recurrent advanced cancer and concomitant COVID-19 infection, conservative treatment was carefully continued, and the patient’s symptoms gradually improved. The thoracic cavity drains were then removed, and no recurrence or exacerbations were observed. Thus, conservative treatment was effective for mediastinal and subcutaneous emphysema, which occurred as complications of pneumothorax ex vacuo in our patient.

While our patient developed pneumothorax ex vacuo after drainage of malignant pleural effusion, similar pathological conditions have been reported as complications of hepatic hydrothorax in patients with liver cirrhosis [16]. In patients with liver cirrhosis, managing sodium and body fluids is difficult and ascitic fluid accumulates. The ascitic fluid passes from the abdomen into the chest through the fine pores of the diaphragm, which results in exudative pleural effusion. Consequently, a pneumothorax can occur. Hepatic hydrothorax is caused by fibrosis and the scarring of the visceral pleura due to chronic inflammation caused by pleural effusion. It appears to occur through a mechanism similar to that of pneumothorax ex vacuo [17]. Recently, a case of pneumothorax after COVID-19 was reported, followed by a re-expansion of pulmonary edema after drainage [14].

In the future, as the prevalence of COVID-19 continues to increase worldwide, the risk of pneumothorax may also increase. Our experience with this case suggests that sufficient careful consideration is required for thoracic cavity drainage for malignant pleural effusion and the retention of other fluids because chronic inflammation in the thoracic cavity is involved in the onset of pneumothorax ex vacuo.

## 4. Conclusions

In the case presented herein, pneumothorax ex vacuo was developed after pleural drainage for malignant pleural effusion. An active COVID-19 infection was also considered as an etiology. Although the prognosis of this case was thought to be poor, conservative treatment with thoracic drainage rather than surgery was remarkably effective for extensive mediastinal and subcutaneous emphysema caused by pneumothorax ex vacuo. With the spread of COVID-19, the number of affected patients is expected to increase significantly, and the risk of pneumothorax due to lung fragility is also expected to increase. Therefore, drainage for malignant pleural effusion and other fluid retention needs to be carefully considered according to the condition of each patient.

## Figures and Tables

**Figure 1 medicina-59-00709-f001:**
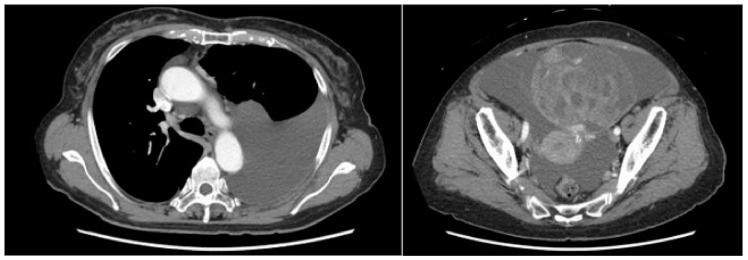
Retention of pleural and ascitic fluid was observed.

**Figure 2 medicina-59-00709-f002:**
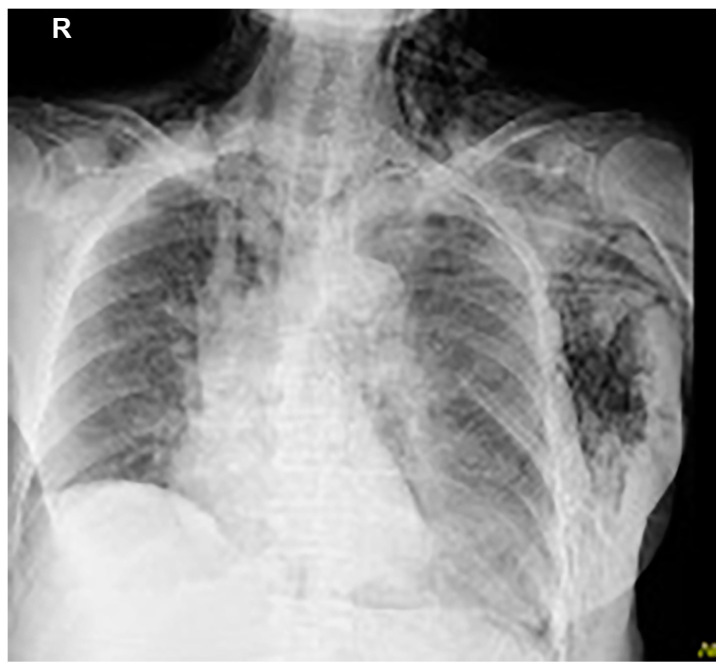
Mediastinal and subcutaneous emphysema was observed in the anterior mediastinum and on the left sides of the chest, respectively.

**Figure 3 medicina-59-00709-f003:**
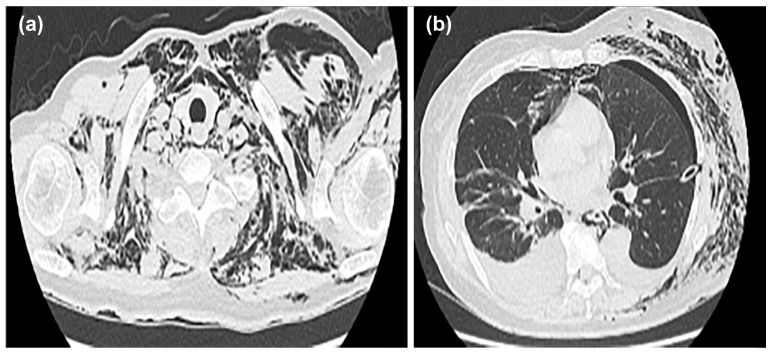
Mediastinal emphysema was observed in the anterior mediastinum, and subcutaneous emphysema was predominant on the left side of the chest (**a**). A drain was inserted between the lobes in the left thoracic cavity (**b**).

**Figure 4 medicina-59-00709-f004:**
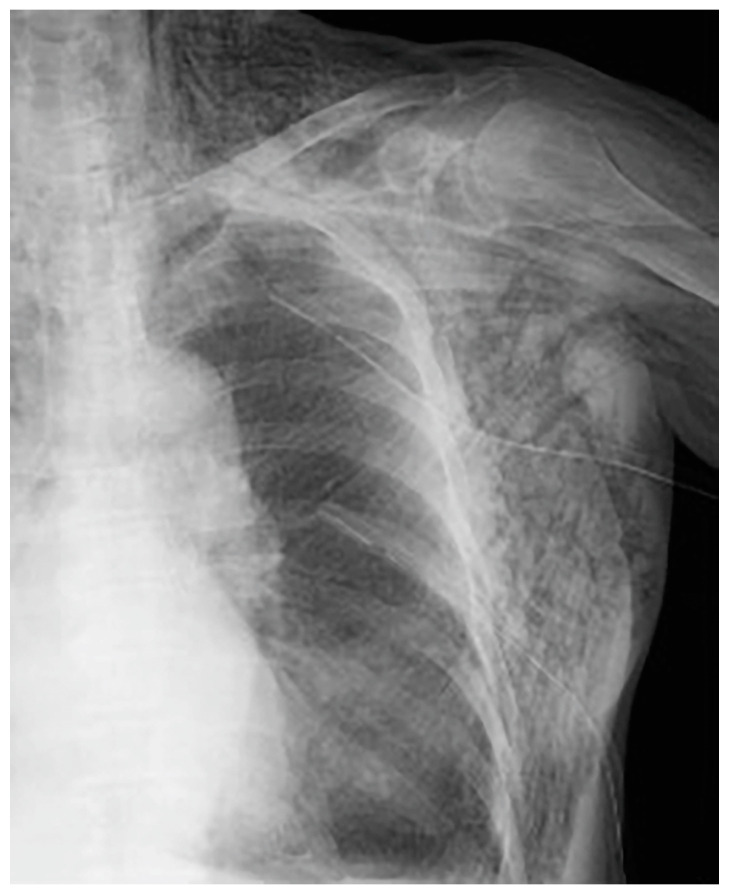
An additional drain (20 Fr) was placed from the left anterior chest to the left thoracic cavity.

**Figure 5 medicina-59-00709-f005:**
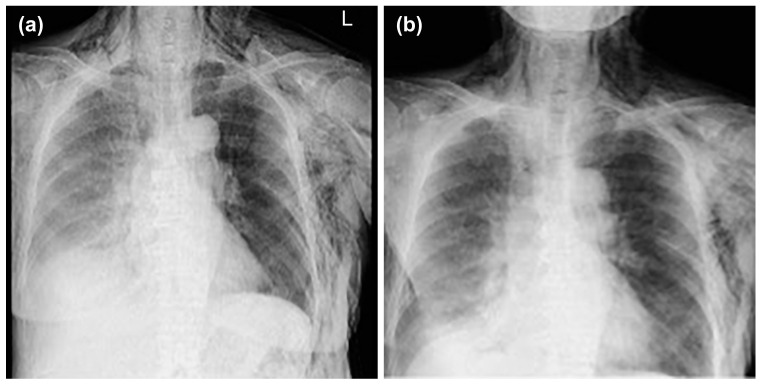
The clinical course after placement of the first thoracic cavity drain. Day 1 (**a**), Day 4 (**b**).

**Figure 6 medicina-59-00709-f006:**
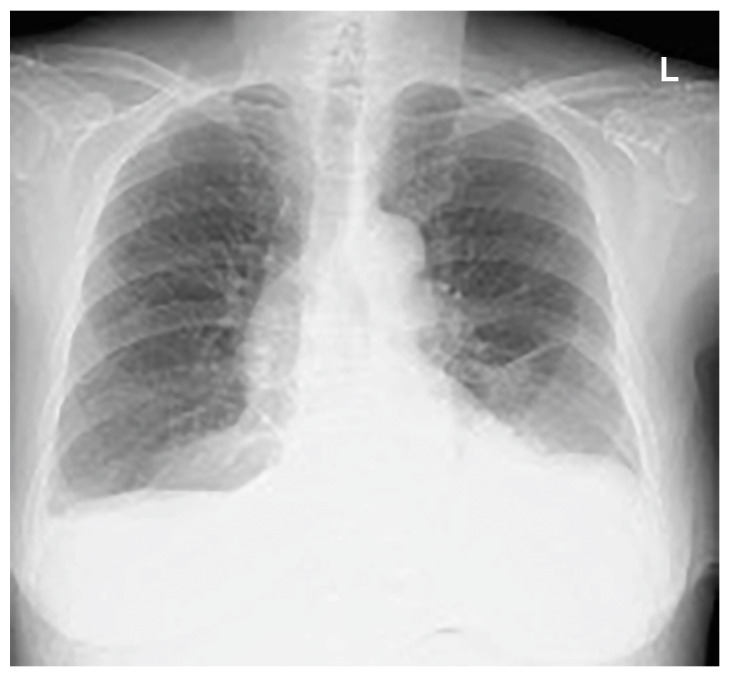
The day before hospital discharge. Mediastinal and subcutaneous emphysema had improved.

## Data Availability

Not applicable.

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
