# Peer review of "A Case of Pneumothorax Ex Vacuo Associated with COVID-19"

_medicina, 2023, doi:10.3390/medicina59040709_

Round 1

Reviewer 1 Report

Dear Authors, I carefully read yours case report.

In my opinion the authors should describe better the correct timing to perform the thoracentesis, because is not very clear. Indeed, I do not understand whether the drain was initially placed for the only mediastinal and subcutaneous emphysema (how it appears in Figure 1) or for the pleural effusion (how it appears in Figure 2). 

Author Response

Thank you for this comment.   You are correct; we agree that the chronological description of the case presentation was very confusing and have made significant modifications accordingly.

Reviewer 2 Report

This case report presents a rare thoracentesis complication that poses a  challenge to it's management. Since there are few cases of ex vacuo pneumothorax described in the recent literature, any encountered case is worth presenting.

However, the manuscript would benefit of a more clear narative. The timing and reason of thoracic drainage is confusing and differs between the abstract and the manuscript. 

The introduction section could provide more insight into the topic of ex vacuo pneumothorax. Consider rewriting it.

The discussions should address aspects specific to ex vacuo pneumothorax and more comprehensive analyse regarding it’s management.

The conclusions duplicate the last paragraph of discussions. Consider rewriting it.

Here are more recent published articles on the same topic, the author might consider:

Venkitakrishnan, Rajesh; Augustine, Jolsana; Ramachandran, Divya; Cleetus, Melcy. Pneumothorax ex vacuo: Three cases of an uncommon entity. Lung India 40(2):p 169-172, Mar–Apr 2023. | DOI: 10.4103/lungindia.lungindia_517_22

Saha BKHu KShkolnik B Non-expandable lung: an underappreciated cause of post-thoracentesis basilar pneumothorax

Florecki K, Anaokar J, Katlic M, Carter Y (2017) Pneumothorax Ex Vacuo Following Thoracentesis for Persistent Pleural Effusion. J Surg Clin Pract 2:1 

Author Response

This case report presents a rare thoracentesis complication that poses a  challenge to it's management. Since there are few cases of ex vacuo pneumothorax described in the recent literature, any encountered case is worth presenting.

  1. However, the manuscript would benefit of a more clear narative. The timing and reason of thoracic drainage is confusing and differs between the abstract and the manuscript.

Answer: Thank you for this comment. You are correct; we agree that the chronological description of the case presentation was very confusing and have made significant modifications accordingly.

  1. The introduction section could provide more insight into the topic of ex vacuo pneumothorax. Consider rewriting it.

Answer: As you have pointed out, the background information was lacking. Accordingly, we have added relevant text to the introduction section.

  1. The discussions should address aspects specific to ex vacuo pneumothorax and more comprehensive analyse regarding it’s management.

Answer: Thank you for pointing this out. We have added text to the discussion accordingly, along with additional citations to new references per the advice provided by the reviewer in item 5 below.

  1. The conclusions duplicate the last paragraph of discussions. Consider rewriting it.

Answer: We apologize for this mistake. We have adjusted the wording and content of this section.

  1. Here are more recent published articles on the same topic, the author might consider:

Venkitakrishnan, Rajesh; Augustine, Jolsana; Ramachandran, Divya; Cleetus, Melcy. Pneumothorax ex vacuo: Three cases of an uncommon entity. Lung India 40(2):p 169-172, Mar–Apr 2023. | DOI: 10.4103/lungindia.lungindia_517_22

Saha BK, Hu K, Shkolnik B Non-expandable lung: an underappreciated cause of post-thoracentesis basilar pneumothorax BMJ Case Reports CP 2020;13:e238292.

Florecki K, Anaokar J, Katlic M, Carter Y (2017) Pneumothorax Ex Vacuo Following Thoracentesis for Persistent Pleural Effusion. J Surg Clin Pract 2:1

Answer: Thank you for providing these sources. All three references have been cited in the text and reference list. We also consulted them when considering the composition of the minutes.

Round 2

Reviewer 2 Report

Thank you to the authors for revising the manuscript. They did a great job, however I would recommend to look further into the narrative of the article and clarify the sequence of events. L 69 states  that thoracentesis was performed. Can you please explain if it was done just for extracting fluid for tests or was it any amount evacuated? L 73 '' a large left pleural effusion was determined...'' if thoracentesis was performed earlier, than discussion is needed to explain the finding.

Please clarify also the timing of the two drains. L86 ..''a drain was placed...''

L95,96 ..''a drain was then inserted..''

L 102...''an additional drain..''

Author Response

・Thank you to the authors for revising the manuscript. They did a great job, however I would recommend to look further into the narrative of the article and clarify the sequence of events

Answer: Thank you for your time. We agree with all of your points. I will respond to each of your points below.

・L 69 states  that thoracentesis was performed. Can you please explain if it was done just for extracting fluid for tests or was it any amount evacuated?

Answer: Each drained 1000 ml for testing. This has been noted in the text.

・L 73 '' a large left pleural effusion was determined...'' if thoracentesis was performed earlier, than discussion is needed to explain the finding.

Answer: About two weeks after the thoracentesis for the first examination, a large amount of pleural fluid accumulated again. I am sorry that its progress is very difficult to understand. We have corrected it.

・Please clarify also the timing of the two drains. L86 ..''a drain was placed...''

Answer: As you said, I think the wording was very confusing. After the appearance of subcutaneous emphysema, a thoracic drain was inserted. After that, a CT scan was taken. On that day, only one drain was inserted. The next day, an additional second drain was placed because the x-ray showed an exacerbation trend. Corrected text. We are very sorry.

・L95,96 ..''a drain was then inserted..''

Answer: Deleted.

・L 102...''an additional drain..''

Answer: The wording in the text has been changed. We appreciate your understanding.